# Functional Characterization of ShK Domain-Containing Protein in the Plant-Parasitic Nematode *Bursaphelenchus xylophilus*

**DOI:** 10.3390/plants13030404

**Published:** 2024-01-30

**Authors:** Madalena Mendonça, Cláudia S. L. Vicente, Margarida Espada

**Affiliations:** MED—Mediterranean Institute for Agriculture, Environment and Development & CHANGE—Global Change and Sustainability Institute, Institute for Advanced Studies, and Research, Universidade de Évora, Pólo da Mitra, Ap. 94, 7006-554 Évora, Portugal; maria.silva@uevora.pt (M.M.); cvicente@uevora.pt (C.S.L.V.)

**Keywords:** molecular plant–nematode interaction, parasitism, pinewood nematode, oxidative stress

## Abstract

ShK domain-containing proteins are peptides found in different parasitic and venomous organisms. From a previous transcriptomic dataset from *Bursaphelenchus xylophilus*, a plant-parasitic nematode that infects forest tree species, we identified 96 transcripts potentially as ShK domain-containing proteins with unknown function in the nematode genome. This study aimed to characterize and explore the functional role of genes encoding ShK domain-containing proteins in *B. xylophilus* biology. We selected and functionally analyzed nine candidate genes that are putatively specific to *B. xylophilus*. In situ hybridization revealed expression of one *B. xylophilus* ShK in the pharyngeal gland cells, suggesting their delivery into host cells. Most of the transcripts are highly expressed during infection and showed a significant upregulation in response to peroxide products compared to the nematode catalase enzymes. We reported, for the first time, the potential involvement of ShK domain genes in oxidative stress, suggesting that these proteins may have an important role in protecting or modulating the reactive oxygen species (ROS) activity of the host plant during parasitism.

## 1. Introduction

Plant pathogens cause severe economic and ecological problems in a wide range of crops and forestry plant species in Europe [1]. The migratory plant-parasitic nematode *Bursaphelenchus xylophilus* (pinewood nematode, PWN) is a European A2 quarantine organism that causes economic damage to the forestry industry [2,3]. Pine wilt disease is caused by the PWN, which was introduced in the early 20th century into Asian countries [4]. Since then, many native tree species, mainly from the genus *Pinus* spp. (pine trees), from both European (Portugal; Spain) [5,6] and Asian (Japan, Korea, China, and Taiwan) [7] forests were considered highly susceptible to PWN.

The interaction between plant-parasitic nematodes (PPN) and their hosts is mediated by parasitism-related proteins (named effectors), which are secreted proteins, with different functions, delivered into the host for the successful infection by the nematode [8]. Due to advances in genomics and transcriptomics, several proteins of the *B. xylophilus* have been identified as important during the interaction with the host, which includes different cell-wall degrading proteins, detoxification and antioxidant enzymes, and other small molecules that can facilitate nematode survival inside the host tissues, such as toxin peptides, venom-like proteins, and other peptides whose functions are unknown [9,10].

ShK domain-containing proteins are present in both plant (e.g., *Arabidopsis thaliana* and *Oryza sativa*) and animal kingdoms (e.g., *Gallus gallus*, *Xenopus tropicalis,* and *Cryptosporidium parvum*) [11]. The ShK domain-containing protein is characterized by a 35/37-residue peptide toxin, originally found in cnidaria species, sea anemones *Stichodactyla helianthus* [12], and *Bunodosoma granulifera* [13], and described as a potassium channel (K^+^) blocker. Collectively, a large superfamily of proteins that contain domains resembling ShK or BgK are referred to as ShKT domains [11]. The structure of ShK domain-containing proteins is defined by six cysteines, two α-helices, and three disulfide bonds (Cys1–Cys6, Cys2–Cys4, and Cys3–Cys5) [14,15]. These peptide sequences can have single or multiple ShKT domains and can be associated with other domains [16]. Depending on the structural organization of the polypeptide, this domain is mostly associated with metalloproteases, tyrosinases, prolyl-4-hydroxylases, oxydoreductases, and peroxidases, which can be combined with various additional domains (e.g., epidermal growth factor-like domains, trypsin-like serine protease, or thrombospondin-type repeats) [11]. A large number of multidomain proteins from both plants and animals, like polychaets [17], marine annelids [18], and sea snails [19], have the same evolutionary highly conserved protein motif defined by the typical structural fold of ShKT [11]. The ShKT domain-containing proteins in the phylum Nematoda are present in the free-living *Caenorhabditis elegans*, with ca. 66 secreted ShKT proteins containing multiple copies of the ShK toxin domain [20], in the entomopathogenic nematode *Steinernema carpocapsae* [21,22], in the plant-parasitic nematodes *Radopholus similis* [23], *Pratylenchus penetrans* [24], *Meloidogyne javanica* [25], *Heterodera schachtii* [26], and in the animal-parasitic species *Toxocara canis* [15] and *Trichostrongylus colubriformis* [27]. In the parasitic and venomous species, the peptides with the ShKT domain have been shown to function as K^+^ channel blockers [15,28,29,30]. In the phylum Nematoda, ShK toxins present in the animal-parasitic nematodes *Ancyclostoma caninum* (AcK1) and *Brugia malayi* (BmK1) are used as a selective inhibitor of K_v_ channels to treat autoimmune disease [30]. In the entomopathogenic nematode *S. carpocapsae*, the presence of ShK peptides has been associated with the modulation of the host immunity during infection and their potential to be an insecticidal peptide [21,22]. In the plant-parasitic nematode *M. javanica*, ShK peptides (*MjShKT*) are highly secreted into the host cells and inhibit programmed cell death, allowing the nematode to invade and reproduce inside the host tissues [25]. K^+^ channels are known to be indirectly engaged in plant defense responses through the action of reactive oxygen species (ROS) (biotic and abiotic stress responses), indicating that the ShK toxin may have a similar impact on K_v_ channels in plants [31,32].

The ShKT domain is a broad and diverse family present in plants and animals. Among parasitic nematodes, few studies have been performed with these proteins. In PPNs, although ShK domain-containing protein is present, their function(s) still remains unclear in the biology of the nematodes. Understanding their role may increase our knowledge of how PPNs are able to modulate host plants during infection. Based on the previous transcriptomic data, this study aims to characterize the functional role of ShK domain-containing proteins that are putatively specific to *B. xylophilus* and are highly expressed during host infection. We report a new putative role for ShK proteins and suggest a new target gene(s) for the molecular PWN control.

## 2. Results

### 2.1. Characterization of ShK Domain-Containing Protein from B. xylophilus

The analysis of both *B. xylophilus* genome and transcriptome [9,33,34] identified 96 transcripts potentially identified as ShK domain-containing proteins with unknown functions. From them, 64% of the transcripts that encode for a ShK domain-containing protein (corresponding to 62 coding sequences) have a predicted signal peptide, suggesting their involvement in the secretory pathway of PWN. Similar to other organisms, the *B. xylophilus* ShK domain-containing proteins are composed of 35–37 amino acid residues and contain six conserved cysteine residues [12], characteristic of the ShKT motif. *B. xylophilus* transcripts potentially encoding the ShK domain-containing proteins consist of one or more ShKT domains (IPR003582), or ShKT domains combined with other domains (Figure 1A) [11]. A previous study showed that ShK domain-containing protein architecture displays several structural positions depending on whether it is a single or multiple ShKT domain, or whether it is associated with another domain [19]. From the 96 transcripts, 25 were predicted with a single ShKT domain, 37 with a multi ShKT domain (between 2 and 4), and 34 transcripts were combined with other domains (Figure 1B). The presence of different domain architectures may suggest that these range of proteins play different functions in the nematode. In this nematode, transcripts are associated with several domains, such as metallopeptidases M12 (IPR001506) and M14 (IPR000834), haem peroxidase (IPR019791), tyrosinase copper-binding domain (IPR002227), epidermal growth factor (EGF)-like domain (IPR000742), CUB domain (IPR000859), and others.

BLASTp sequence analysis (e-value cutoff of 1 × 10^−5^ and a bitscore >50) was used to identify significant matches to potential orthologs/homologs of *B. xylophilus* in Wormbase ParaSite and NCBI non-redundant databases. The results revealed that from the total of 96 transcripts of *B. xylophilus*, 70 had a high degree of identity (50–98%) with proteins from other PPNs as well as animal-parasitic and free-living nematodes, and 26 transcripts had similarity with proteins from *Bursaphelenchus* spp. Furthermore, 15 transcripts had significant sequence similarity only to *B. xylophilus* and no match with the nematodes of different genera or other organisms, suggesting that these are *B. xylophilus* putatively specific transcripts. From these 15 ShK domain-containing proteins, 9 were selected for analysis based on the existence of a predicted signal peptide in the N-terminal and the presence of only the ShKT domain, without being associated with other domains (Figure 2A; Table 1). The sequences of the transcripts of interest in this study are BXY_0074200, BXY_0728300, BXY_1389600, BXY_0192800, BXY_0841100, BXY_1304100, BXY_1328700, BXY_1389700, and BXY_0654300. A common characteristic of these nine transcripts is the fact that they all have two ShKT domains (D1 and D2).

### 2.2. In Silico Analysis and Phylogeny of the Nine Candidate B. xylophilus Putatively Specific Transcripts

In silico analysis of the candidate genes revealed that the full-length DNA sequence ranged between 502 and 3250 bp, with 1 to 4 introns, and the full-length cDNA was between 423 bp and 2625 bp. The translation of the full-length cDNA revealed protein sequences between 140 and 874 amino acids (Table 1).

The expression levels of these nine transcripts were predicted from RNA-seq data generated in a previous study [9], and the fold change for each transcript was calculated according to host infection (6 and 15 days post-infection) compared with the non-parasitic stage. Most of these transcripts were highly expressed (fold change, FC > 1.5) (Figure 2B), except for genes BXY_1389700 and BXY_0841100 (FC < 1) that have a lower expression and BXY_1328700 that has no expression during infection. For these secreted *B. xylophilus* ShK domain-containing sequences, full-length amplifications from the genomic DNA and cDNA confirmed the in silico prediction of the selected genes and transcripts.

To perform the phylogeny analysis, we aligned the sequence domains of the nine candidate transcripts putatively specific to *B. xylophilus* with sequence domains from other nematodes and plants (Appendix A). Therefore, a maximum-likelihood (ML) (Figure 3) and a neighbor-joining (Appendix A) phylogeny trees were constructed. The multiple sequence alignment revealed that ShKT D1 domain sequences were more similar to each other than the ShKT D2 domain. The same pattern was also observed within ShKT D2 domain sequences. The ML tree showed two clades: one with ShKT domain sequences of plants, considered an outgroup (*A. suecica* KAG7546197 and *Eucalyptus grandis* XP_010033232); and the other clade with ShKT domain sequences from Nematoda. In this last clade, several sub-clades are formed. The first sub-clade is composed of animal-parasitic species *B. malayi* (VIO92324) and *T. canis* (KHN87551), and free-living nematodes *B. okinawaensis* (CAD5214648) and *Diploscapter pachys* (PAV73547) (domain 1). The second sub-clade presents several clusters, from which the first is formed with domain 2 of the same species previously shown in the first sub-clade. In fact, the full protein sequences of these two sub-clades are associated with other domains, such as the di-copper center-containing domain and the tyrosinase domain. Another cluster in the second sub-clade is grouped into the nine *B. xylophilus* domains with *Strongyloides ratti* (CEF69303) (animal-parasitic) and *C. elegans* (CCD70350) (free-living) (D1 and D2). Also, this cluster showed *M. incognita* (Minc3s00855g18130) and *H. glycines* (Hetgly.G000014452) (plant-parasitic) with a significant bootstrap (higher than 90%) sharing a single ShKT domain. Some candidate genes (BXY_0192800 D1, BXY_0074200 D1, BXY_1389600 D1, BXY_1389700 D1, and BXY_0654300 D1) have a low bootstrap (<40%); therefore, it is not possible to infer their relation in the tree. However, BXY_1304100 D1 and BXY_1328700_D1 have a significant bootstrap (89%). These data may suggest that these genes putatively specific to *B. xylophilus* could be obtained independently from the others, and they do not evolve as a result of duplication of a single ancestor. 

### 2.3. Spatial Expression of the Candidate ShK Domain-Containing Proteins

In situ hybridization is often used to determine the spatial expression of the transcripts in the nematode tissues. We have studied the spatial expression patterns of the nine putatively ShK domain-containing secreted proteins in mixed life-stage nematodes. Positive labeling with the *anti-sense* probe localized one gene in the pharyngeal gland cells tissues (BXY_0728300) (Figure 4A), four genes in the intestine region (BXY_0654300, BXY_1389600, BXY_0074200, and BXY_0192800) (Figure 4B–E), and two genes in the nervous system region (BXY_0841100 and BXY_1389700) (Figure 4). No signal was observed for the two remaining genes (BXY_1328700 and BXY_1304100) and *in sense* probes were used as control. These results suggest that the ShK domain-containing proteins are expressed in different tissues of the nematode, indicating that they may have diverse functions.

### 2.4. High Gene Expression of B. xylophilus ShK Domain-Containing Proteins under Oxidative Stress

The expression profile of the nine transcripts was validated via qRT-PCR (quantitative reverse-transcriptase PCR) under an oxidative stress test to understand their response to the presence of H_2_O_2_. The hydrogen peroxide was intended to represent the ROS products. The specificity and efficiency of all the designed primers were demonstrated in a qRT-PCR amplification for each primer pair that had a single expected size, and the melting curves showed a single peak in the reaction. Relative expression levels for the candidate genes were compared with *B. xylophilus* catalases, *Bxy-ctl-*1 (BXY_1386500), and *Bxy-ctl-*2 (BXY_1745800) [35] since their biochemical function is to catalyze the reaction by which the H_2_O_2_ is decomposed. The results were normalized for the *B. xylophilus* reference gene, *Bx-actin*. At 15mM H_2_O_2_, six candidate genes were significantly upregulated (ANOVA, Tukey’s test; *p* < 0.05), except BXY_0074200, which was downregulated, and BXY_0192800 and BXY_0841100 had no significant expression (*p* > 0.05) (Figure 5). At 30 mM of H_2_O_2_, eight candidate genes showed a significant upregulation (ANOVA, Tukey’s test; *p* < 0.05), except for the gene BXY_0192800 (Figure 5). As reported in a previous study [35], both *Bxy-ctl*-1 (BXY_1386500) and *Bxy-ctl-*2 (BXY_1745800) showed a significant expression in the two treatment conditions. These data may suggest that ShK domain-containing proteins may be participating in response to oxidative stress and, therefore, might be important during the interaction with the host.

## 3. Discussion

The ShK domain-containing proteins are distinguished by a conserved and small ShKT domain, distributed across animal and plant kingdoms, involving toxic peptides and bigger multidomain proteins with a variety of functions [11,12,19]. In this study, we have identified 96 transcripts encoding for ShK domain-containing proteins from one of the most damaging PPN, *B. xylophilus* [9,33], and showed that the putative secreted proteins were enriched in these sequences—64% of these sequences encode a predicted signal peptide—which indicates that it might belong to secretory pathways. These proteins have also been found in the secretome of several animal parasitic nematodes (e.g., *Teladorsagia circumcincta*) [36] and other plant-parasitic nematodes, including *B. xylophilus*, *P. penetrans*, *R. similis*, and *M. javanica* [23,24,25,37]. The number of predicted ShK sequences constitutes one of the largest known protein families in Nematoda (e.g., *T. circumcincta*) [16,36], which is also observed in our data from *B. xylophilus*. Moreover, one study observed a life-stage-specific regulation of the expanded gene family of ShK domain-containing proteins in the animal parasitic nematode, *T. circumcincta* [36]; however, for PWN, this differential expression has not been observed for ShK genes [34]. All *B. xylophilus* predicted ShK domain-containing protein sequences showed a similar structure with single or multidomain ShKT or are associated with other protein domains as observed in other organisms (e.g., cnidaria) [38]. The number and variety of potential combinations of domains in these proteins may be an indication of the diversity of functions associated. 

Nine putatively species-specific candidate genes were selected based on the presence of a predicted signal peptide and two ShKT domains (multidomain). In silico analysis of these genes resulted in a lack of similarity to other sequences, indicating that they are specific to this nematode, as they have only been found in this species so far. These genes were putatively found to be highly expressed during infection of the host [9], suggesting that they might be involved in parasitism. Taking this information, we investigated the spatial expression profiles of these candidate transcript genes and confirmed their expression in different nematode tissues. Two gene transcripts were detected in structures of the nervous system (BXY_0841100 and BXY_1389700) and both have a lower expression during infection with the host (FC < 1). The nervous system is composed of major dorsal and ventral nerves along the body of the nematode, nearly associated with the epidermis, containing cell bodies that make and receive synapses with other neurons [39]. Some parasitic-related genes have been localized in nervous system tissues (e.g., *R. similis*) [23]. Curiously, a PWN catalase, which was used as a positive control for our ROS experiment, expressed in a transgenic *C. elegans* (for *B. xylophilus* catalase, *Bxy-ctl-2*, and BXY_1745800), was also localized in the nervous system [35]. Another validated gene BXY_0728300 was localized in the pharyngeal gland cells, important in plant–nematode interactions. Several studies showed the relevance of pharyngeal gland cells as a producer of several proteins that are secreted and potentially related to parasitism. The ShK domain-containing proteins that have also been identified in other plant-parasitic nematodes, such as *P. penetrans* [24], *R. similis* [23], *M. javanica* [25], and *H. schachtii* [26], are also expressed in the pharyngeal gland cells. *M. javanica* secreted ShK protein, *MjShKT,* which is expressed in the dorsal gland cell and upregulated in parasitic stages and manipulates the plant effector-triggered immune response [25]. In addition, some genes have been expressed in other nematode tissues and still have a role in parasitism. The majority of the ShK-domain encoding genes analyzed showed a signal in the intestine (BXY_0654300, BXY_1389600, BXY_0074200, and BXY_0192800), which may indicate that these genes may be involved or have protective activity against stress-related plant molecules during migration or infection. Previous studies have shown that *B. xylophilus* has several proteins that are highly expressed during infection and broadly expressed in the intestine, such as two catalases [35] and several detoxification enzymes (e.g., glutathione S-transferase) [40]. Therefore, this emphasizes the multilayer strategy of PWN which may have a protective role against plant defense compounds [9].

All eight ShK domain-containing protein genes showed an upregulation in the expression levels in response to the H_2_O_2_ at 30mM treatment, suggesting that these proteins might be responsive to H_2_O_2_ and indirectly involved in the ROS scavenging. We reported, for the first time, the potential involvement of ShK domain genes in oxidative stress. In summary, due to its multidomain architecture and large number, ShK domain-containing proteins may have a diversity of functions in nematodes. Previously described functions of ShK domain-containing proteins in parasitic nematodes, including those involved in immuno-modulatory action [30] or response in order to establish and maintain the feeding sites [25]. Our data suggest that the migratory endoparasitic nematode *B. xylophilus* uses highly secreted ShK domain-containing proteins to respond to ROS products, secreting into the host and at the same time in the digestive and nervous systems. Understanding the role of ShK domain-containing proteins may increase our knowledge of how nematodes modulate hosts during infection and develop new target molecules for nematode control.

## 4. Materials and Methods

### 4.1. Biological Material

The isolate of *B. xylophilus*, isolate HF (Herdade da Ferraria, Setúbal, Portugal), used in this study was cultured at NemaLab, University of Évora. The nematodes were cultured on *Botrytis cinerea* on barley seeds at 25 °C [41] for approximately 10 days. Mixed life-stage nematodes were harvested using the modified Baermann funnel technique [42] for 6 h, followed by sieving (38 μm).

### 4.2. Phylogenetic Analysis of B. xylophilus ShK Domain-Containing Protein Sequences

A comprehensive list of 96 transcripts encoding the ShK domain-containing sequences was identified using the previous *B. xylophilus* genomic [33] and transcriptomic data [9]. Protein sequence similarity analysis was performed by BLASTp [43,44] to analyze the similarity to other amino acid sequences against the NCBI NR (non-redundant) database (https://www.ncbi.nlm.nih.gov/, accessed on 8 August 2023). The homology of the sequences was also analyzed against publicly available nematode genomes at Wormbase ParaSite database (http://parasite.wormbase.org, accessed on 5 September 2023), with phylum Nematoda. All BLASTp searches were carried out with a cutoff of 1 × 10^−5^ and a bitscore of >50. The characterization and prediction of the functional domains of ShK domain-containing proteins were carried out using the InterProScan database (https://www.ebi.ac.uk/interpro/, accessed on 21 September 2023) [45]. Secreted ShK domain-containing proteins were predicted based on the presence of signal peptide (SP) by SignalP (v5.0) [46] and InterProScan. The protein sequences of the ShK domain were downloaded from the UniProtKB database. From the list of 96 transcripts, a subset of 9 candidate genes encoding ShK domain-containing proteins were selected for further analysis due to the lack of similarity with any other organism, indicating that these genes may be *B. xylophilus*-specific. The sequences of the genes of interest in this study are available in v.1.3. of the *B. xylophilus* (BioProject PRJEA64437) [33]. To perform the phylogenetic analysis, multiple sequence alignment was performed using MAFFT (v.7.0) [47], followed by sequence trimming using trimAl (v.1.4.1) [48]. The phylogenetic trees were generated by two different methods: neighbour-joining and maximum-likelihood analysis (PhyML; in CLC Sequence mainWorkbench v.21, Qiagen) supported by 1000 bootstrap replications. Phylogenetic trees were visualized and edited in iTOL (https://itol.embl.de/, accessed on 14 November 2023) [49].

### 4.3. Validation of the Candidate Genes

To validate the nine candidate genes in the *B. xylophilus* genome and transcriptome, primers were designed to amplify full-length sequences by PCR using Supreme NZYTaq II 2× green master mix (NZYTech, Lisbon, Portugal). DNA extraction was performed using the NZY Tissue gDNA Isolation kit (NZYTech, Lisbon, Portugal) according to the manufacturer’s instructions. The PCR reaction was performed with an initial denaturation step for 3 min at 95 °C, followed by 35 cycles of denaturation for 30 s at 94 °C, annealing for 30 s at 55 °C and extension for 30 s at 72 °C, followed by a final extension step of 10 min at 72 °C. The amplified products were visualized in 1.4% agarose gel and were purified using MinElute^®^ PCR or Gel Extraction Kits (QIAGEN, Hilden, Germany) according to the manufacturer’s instructions, and then sent for sequencing (STAB VIDA, Costa da Caparica, Portugal). Sequencing results were analyzed at BioEdit [50].

### 4.4. Oxidative Stress Assay

The ShK domain-containing proteins with predictive expression in *B. xylophilus* were tested under oxidative stress with hydrogen peroxide (H_2_O_2_) in two different concentrations of 15 mM and 30 mM [35]. A 96-well plate was prepared as follows: each well received 50 μL of H_2_O_2_ and 50 μL of mixed life-stage nematode solution (approximately 280 nematodes per 50 μL of sterilized distilled water). A control treatment with distilled water was also prepared. For each treatment condition, three technical replicates per experiment were prepared. After a 24-h incubation at room temperature, the plate was examined under a stereomicroscope (Olympus SZX12) to check for animal viability. Nematodes with no movements, after mechanical stimulation, were considered dead. After a 24 h exposure to H_2_O_2_, most of the nematodes were alive, which were collected, after which we proceeded to RNA extraction.

### 4.5. RNA Extraction and Gene Relative Expression Profile

Total RNA from 24 h stressed *B. xylophilus* was extracted from the three conditions (0 mM, 15 mM H_2_O_2_, 30 mM H_2_O_2_) using the RNeasy Mini kit following the manufacturer’s instructions (QIAGEN, Hilden, Germany). The quantity and quality of the extracted RNA were assessed by an ND-2000/2000c NanoDrop spectrophotometer (Thermo Fisher Scientific, Waltham, MA, USA), and 200 ng of cDNA final concentration was synthesized using the SuperScript^®^ III reverse transcriptase (Invitrogen by Thermo Fisher Scientific, Waltham, MA, USA). Prior to qRT-PCR, primers previously designed using Prime 3 software [51] were tested for specificity (Appendix A). Relative gene expression was analyzed via quantitative RT-PCR (qRT-PCR) using SYBR green assay SensiFAST™ SYBR Lo-ROX mix, 2× (Bioline Reagents Ltd, Meridian Bioscience, London, UK) and performed using the QuantGene 9600 (Bioer Technology, Hangzhou, China). The reaction mix comprised 10 μL of SYBR green assay, 0.8 μL of each 10 mM primer, and 1 μL of cDNA (10 ng/μL) in a final volume of 20 μL. For each treatment, two biological replicates were performed with three technical replicates. Negative controls with no template were added for each qRT-PCR run. The amplification was performed with initial denaturation at 95 °C for 2 min, 39 cycles of denaturation at 95 °C for 5 s, and annealing and extension at 60 °C for 20 s, followed by the melting stage. Primer specificity was confirmed by a single peak at the PCR product’s melting point. Bxy-actin (BXY_0322900) was used as a reference gene for normalization according to the 2^−ΔΔCT^ method [52]. The data were analyzed with Ct (cycle threshold) values in the control treatment (without H_2_O_2_) and oxidative stress conditions (15 mM and 30 mM H_2_O_2_). Statistical analyses were performed with Jamovi software (v.2.3.21) (https://www.jamovi.org) using analysis of variance (ANOVA), and means were compared using Tukey’s test with statistical significance at *p*-value < 0.05.

### 4.6. In Situ Hybridization Assay

To determine the spatial expression patterns of candidate genes, in situ hybridization using digoxigenin (DIG)-labeled probes was performed based on the protocol described by [53]. Total RNA from *B. xylophilus* was extracted, as previously described. cDNA was synthesized using 50 ng of total RNA by the SuperScript^®^ III first-strand synthesis system for RT-PCR (Invitrogen by Thermo Fisher Scientific, Waltham, MA, USA) following the manufacturer’s instructions. For each gene, a specific fragment of approximately 150–250 bp was amplified by PCR. The primers used are shown in Appendix A. The amplification reaction mixture consisted of 1U Platinum^®^ Taq DNA Polymerase High Fidelity (Invitrogen by Thermo Fisher Scientific, Waltham, MA, USA), 1 mM of each primer (forward and reverse), 10 mM of dNTPs, and 50 ng of template cDNA in a 50 μL reaction volume. The PCR reaction was performed with an initial denaturation step for 3 min at 95 °C, followed by 35 cycles of denaturation for 30 s at 94 °C, annealing for 30 s at 55 °C, and extension for 30 s at 72 °C, followed by a final extension step of 10 min at 72 °C. The amplified products were visualized in 1.4% agarose gel and were purified by MinElute^®^ PCR or Gel Extraction Kits (QIAGEN, Hilden, Germany). The resulting PCR products were used as a template for the generation of both sense and antisense DIG-labeled probes using a DIG DNA Labeling Mix (Roche Diagnostics GmbH, Mannheim, Germany). The antisense probe is synthesized with the primer reverse and is complementary to the transcript mRNA sequence of interest, opposite to the sense probe synthesized with the primer forward (which will not hybridize with the mRNA) and used as a negative control. The probe synthesis (each per transcript) was performed with more than 50 ng of amplified specific fragments. A reaction was performed with an initial polymerase activation step for 2 min at 95 °C, followed by 35 cycles of denaturation for 15 s at 95 °C, annealing for 30 s at 50 °C, and extension for 90 s at 72 °C, followed by a final extension step of 5 min at 72 °C. For the mRNA in situ hybridization in the nematode’s tissues, mixed life-stage *B. xylophilus* were fixed overnight in 2% (wt/vol) paraformaldehyde and cut using a vibrating aquarium air pump with a razor blade vertically on the slide, which was moved slowly back and forth across the nematode suspension. Nematodes were pre-treated with proteinase K for 30 min at room temperature in the rotator and hybridized for 16 h at 49 °C, with both sense and antisense probes for each candidate transcript. Then, hybridized nematode tissues were detected using anti-Digoxigenin-AP FAB fragments (Roche Diagnostics GmbH, Mannheim, Germany) conjugated to alkaline phosphatase detection buffer and its substrate, NBT/BCIP Stock Solution (Roche Diagnostics GmbH, Mannheim, Germany). Nematode segments were observed using a light microscope (Olympus BX50), and images were taken using the software Cell^D^ 3.2 (Olympus Soft Imaging Solutions).

## Figures and Tables

**Figure 1 plants-13-00404-f001:**
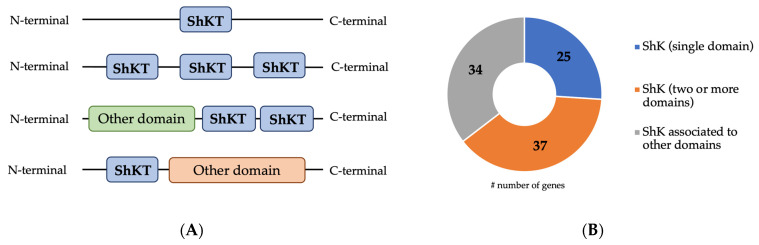
Representation and distribution of the main ShK domain-containing proteins predicted in *B. xylophilus.* (**A**) Representation of the architecture and position of the ShKT domain in ShK domain-containing proteins: one domain ShKT; multidomain ShKT and domain ShKT associated with other domains; (**B**) distribution of ShK domain-containing proteins predicted in the *B. xylophilus* transcriptome.

**Figure 2 plants-13-00404-f002:**
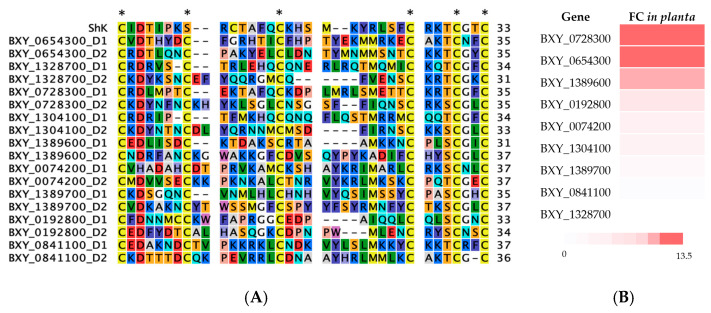
Characterization of the nine candidate ShK domain-containing proteins (*B. xylophilus* genes): (**A**) multiple sequence alignments with domain 1 (D1) and 2 (D2), and the ShKT sequence domain originally from the sea anemone *Stichodactyla helianthus*, including the six conserved cysteine residues (indicated by an asterisk) which characterize the ShKT domain; (**B**) representation of the ratio of the average normalized expression from the nine candidate genes, ranked by their fold change values (FC). Fold change ratio refers to the expression during infection of the host (6 and 15 days post-infection) compared with a non-parasitic stage.

**Figure 3 plants-13-00404-f003:**
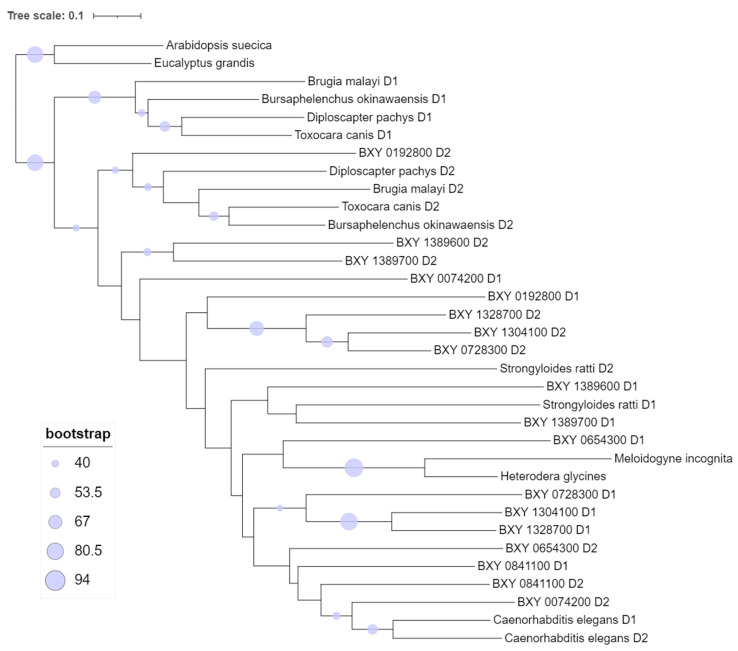
Maximum-likelihood (ML) phylogenetic tree that represents the protein sequence similarity between the nine candidate genes putatively specific to *B. xylophilus* predicted ShK domain-containing proteins and other organisms, such as plant-parasitic, animal-parasitic, and free-living nematodes and plants, as well as ShK domain-containing proteins with domain 1 and domain 2.

**Figure 4 plants-13-00404-f004:**
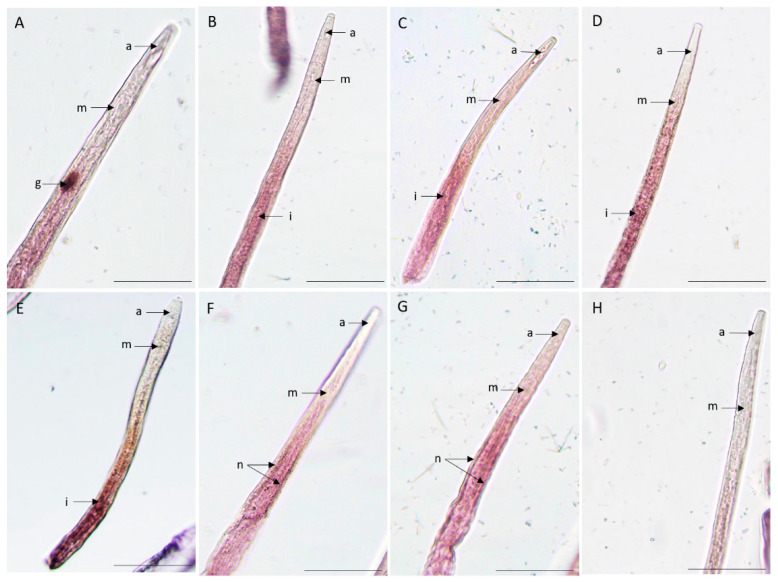
Detection of nine candidate genes, ShK domain-containing proteins, and *B. xylophilus* transcripts by in situ hybridization. Transcripts were localized in different tissues of the nematode using the complementary anti-sense DIG-labeled probes. (**A**) BXY_0728300, (**B**) BXY_1389600, (**C**) BXY_0074200, (**D**) BXY_0192800, (**E**) BXY_0654300, (**F**) BXY_0841100, (**G**) BXY_1389700, and (**H**) sense probe (control). The different nematode anterior sections of the body are labeled as follows: a: anterior region; g: pharyngeal gland cells; i: intestine region; m: median bulb; n: nervous system. Bars = 20 μm.

**Figure 5 plants-13-00404-f005:**
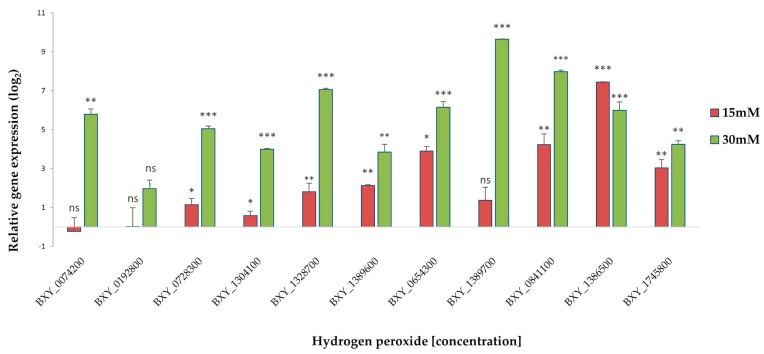
Normalized expression ratio (log_2_) of the nine candidate genes, *Bxy−ctl−1* (BXY_1386500), and *Bxy−ctl−2* (BXY_1745800) under oxidative stress conditions, 15 mM and 30 mM H_2_O_2_, compared to control (0mM). Symbols on the top of the columns indicate statistical significance at (*) *p* < 0.05, (**) *p* < 0.01, and (***) *p* < 0.001 (Tukey’s test), which were significantly different compared with the control treatment (zero). n.s.: not significant.

**Table 1 plants-13-00404-t001:** Characterization of the nine candidate genes from *B. xylophilus* selected in the present study which encode ShK domain-containing proteins. All the genes have only sequence similarity to the species *B. xylophilus* and no other organisms. These genes encode for two ShK domain-containing proteins and have the presence of a signal peptide (SP). Gene length (GL), number of introns (IN), transcript length (TL), and protein size (PS) were predicted for each candidate gene. Sequence similarity and percentage of homology (%) were predicted with BLASTP (the closest homolog in the nr database is indicated by the NCBI accession number).

Gene ID	GL *	IN	TL *	PS **	SP	Sequence Similarity ID (NCBI Accession Number)	BLAST e-Value
BXY_0192800	630	2	510	169	Yes, 1−17	[CAD5228099.1] *Bursaphelenchus xylophilus*, 100%	4 × 10^−123^
BXY_0728300	502	1	429	142	Yes, 1−16	[CAD5230694.1] *Bursaphelenchus xylophilus*, 100%	6 × 10^−94^
BXY_1389600	534	1	492	163	Yes, 1−21	[CAD5234029.1] *Bursaphelenchus xylophilus*, 100%	1 × 10^−111^
BXY_0074200	447	0	447	148	Yes, 1−22	[CAD5228719.1] *Bursaphelenchus xylophilus*, 100%	3 × 10^−105^
BXY_0841100	1170	2	669	229	Yes, 1−17	[CAD5208706.1] *Bursaphelenchus xylophilus*, 100%	8 × 10^−157^
BXY_1304100	922	1	423	140	Yes, 1−16	[CAD5233070.1] *Bursaphelenchus xylophilus*, 100%	2 × 10^−98^
BXY_1328700	673	1	444	147	Yes, 1−23	[CAD5233069.1] *Bursaphelenchus xylophilus*, 100%	3 × 10^−99^
BXY_0654300	3250	4	1926	641	Yes, 1−20	[CAD5235767.1] *Bursaphelenchus xylophilus*, 100%	<0
BXY_1389700	3240	2	2625	874	Yes, 1−21	[CAD5234030.1] *Bursaphelenchus xylophilus*, 100%	<0

* basepairs; ** aminoacids.

## Data Availability

Data are contained within the article and Appendix A.

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
