# Peer review of "Functional Characterization of ShK Domain-Containing Protein in the Plant-Parasitic Nematode Bursaphelenchus xylophilus"

_plants, 2024, doi:10.3390/plants13030404_

Round 1
Reviewer 1 Report
Comments and Suggestions for Authors
Dear Authors,
Your study is a good example of data mining’s impact in biology. Due to the great importance of pinewood nematode, and novelty of your work, I believe your research can be greatly appreciated by researchers who are interested in bioinformatics and plant pathology. The results can also draw attention to those who are working on effectors in nematode-plant interaction. However, I think your paper needs to be reconsidered for some serious downsides.
Here are my main concerns about the manuscript.
1. Phylogenetics: this section is the major conflict and downside in your paper.
- What is the purpose of phylogenetic analysis in this study?
- How does this part add quality to your work?
- Where is the N-J tree result?
- Why did you use plants outgroup for animal ingroup?
- Why did you use 9 different genes domain in a single alignment in a cluster with some other different genes domain from some other nematode? To my opinion, it’s telling you nothing that is why the genes are positioned randomly all over the tree.
2. In discussion and almost all over the manuscript you claimed that the final 9 genes are species-specific. Not having hit from BLASTp is not necessarily proof of that. This study is far away from this claim.
You may also find the file enclosed containing detailed corrections and suggestions. I left some corrections/suggestions regarding the above-mentioned concerns and some others.
I hope these suggestions can benefit you.
Cheers,
Reviewer

Author Response
Dear Reviewer 1,
We enclose a revised version of the manuscript entitled” Functional characterization of ShK domain-containing protein in the plant-parasitic nematode Bursaphelenchus xylophilus” which has been adjusted to take count of your comments.
We have carefully taken into consideration all comments and modified the manuscript text, figures following your suggestions. The responses to your comments are below.
Phylogenetics: this section is the major conflict and downside in your paper.
- What is the purpose of phylogenetic analysis in this study?
- How does this part add quality to your work?
- Why did you use plants outgroup for animal ingroup?
- Why did you use 9 different genes domain in a single alignment in a cluster with some other different genes domain from some other nematode? To my opinion, it’s telling you nothing that is why the genes are positioned randomly all over the tree.
Response: The importance of phylogeny analysis is to understand the evolution and relationship between different organisms that share ShK domain-containing proteins, as well as possible horizontal gene events between PPN and other organisms, as already proved for other proteins (Danchin, 2011, What Nematode genomes tell us about the importance of horizontal gene transfers in the evolutionary history of animals; DOI 10.4161/mge.18776). We show, for the first time, that ShK domains sequences of nematodes cluster according to domain 1 and domain 2. An updated figure 3 was added to the manuscript. Although not shown in this figure, we have used other ShK sequences domains (such as sea anemones, algae, etc) for the construction of the phylogeny tree. However, the resultant tree was not robust enough to draw any conclusions. Concerning the plant sequences for an outgroup, the literature cited in the introduction indicates that “ShK domain-containing proteins are distributed from both plant and animal kingdoms…” (lines 44-45). Based on this, we have used two sequences containing SHK domain from plants to understand the similarity between plants and plant-parasitic nematodes’ sequences.
- Where is the N-J tree result?
Response: A N-J analysis has been provided in supplementary information (Supplementary Figure S1). The same topology of the N-J tree is shown in the ML tree.
In discussion and almost all over the manuscript you claimed that the final 9 genes are species-specific. Not having hit from BLASTp is not necessarily proof of that. This study is far away from this claim.
Response: The sequence similarity analysis was performed with different databases from the NCBI non redundant to Nematoda database from Wormbase Parasite. So far, from what is publicly available to date, this was the most comprehensive result. To clarify this, we have made some edits throughout the text and replaced them with “putative specific genes” (e.g. line 126).
Line 23: Response: we apologize for the inconsistency. We have removed the “ShK domain containing protein” from the keyword section.
Lines 31-32: The cited reference do not state origin of PWN. what is "XX"
Response: The reference to the origin of the PWN, was removed from the main text since the paper does not reference this observation. The sentence was rephrased to: “Pine wilt disease is caused by the PWN, which was introduced in the early 20th century into Asian countries.”
Line 39: rephrase this statement. Now it sounds wrong.
Response: We have made edits to the text that should clarify the sentence: “The interaction between plant-parasitic nematodes (PPN), with their hosts are mediated by parasitism-related proteins (named effectors), which are secreted proteins, with different functions, delivered into the host, for the successful infection of the nematode.”
Line 174: not italic
Repsonse: We have rephrased as requested in the text.
Line 186: use phylogeny instead
Response: We have rephrased as requested throughout the text.
Line 188: Why plants?
Response: As explained above, the literature cited in the introduction indicates that “ShK domain-containing proteins are distributed from both plant and animal kingdoms…” (lines 44-45). Based on this, we have used two sequences containing SHK domain from plants to understand the similarity between plants and plant-parasitic nematodes’ sequences. In different studies there is evidence that plant-parasitic nematodes acquire proteins from other organisms, such as bacteria or fungi.
Line 190: what do you mean by vice versa here?
Response: Minor adjustments have been made to the sentence to clarify it: “The multiple sequence alignment revealed that ShKT D1 domain sequences were very similar to each other. The same occurred with ShKT D2 domain sequences.”
Line 196-197: rephrase the sentence! It is obscure
Response: Minor adjustments have been made to the sentence to clarify it: “In fact, the full protein sequences of this sub-clade are associated with other domains, such as di-copper centre-containing and tyrosinase domains.”
Line 253: all?
Response: The sentence was rephrased to: “At 30mM of H2O2, eight candidate genes showed a significant upregulation (ANOVA, Tukey’s test; p< 0.05), except for the gene BXY_0192800 (Figure 5).”
Line 255: confusing!!
Response: This is explained previously in the text (lines 247-248) and should be clarified following the changes made in the text: “As expected in a previous study [35], both Bxy-ctl-1 (BXY_1386500) and Bxy-ctl-2 (BXY_1745800) showed a significant expression in the two treatment conditions.”. Following the work on Vicente et al., 2015, B. xylophilus catalases can be used as a positive control in oxidative assays, since their biochemical function is to catalyze the reaction by which the hydrogen peroxide is decomposed.
Line 288: Not necessarily!
Response: The sentence was rephrased to: “Nine candidate putative species-specific genes were selected based on the presence of a predicted signal peptide and two ShKT domains (multi-domain).”
Line 304: Are pharyngeal gland cells tissue, in nematodes?
Response: We apologize for the inconsistency. The sentence was rephrased to: “Another validated gene BXY_0728300 was localized in the pharyngeal gland cells, important in plant-nematode interactions.”
Line 312: showed instead of gave
Response: We agree and have rephrased as requested in the text.
Line 320: what about "e BXY_0192800"?!
Response: The sentence was rephrased to: “All eight ShK domain-containing protein genes showed an upregulation in the expression levels in response to the H2O2 at 30mM treatment…”.
Line 346: in silico ?? and ..............
Response: The sentence was rephrased to: “In silico analysis and phylogeny of B. xylophilus ShK domain-containing protein sequences”.
Line 397: How did you defined the viability? and where are the results of this part?
Response: After the oxidative stress assay was performed, the nematodes’s viability was observed under the microscope. Nematodes with no movements, after mechanical stimulation, were considered dead. Since almost all nematodes were alive, all were collected to proceed to the RNA extraction. To make it clearer, the following sentence was added to the text “Nematodes with no movements, after mechanical stimulation, were considered dead. After a 24h-exposure to H2O2, most of the nematodes were alive, collected and we proceeded to RNA extraction.”
Line 411: is it biological replicate?
Response: For each condition in the qRT-PCR, we have used two biological replicates with three technical replicates in each. The text has been edited to make this clear: “For each treatment, two biological replicates were performed with three technical replicates.”
Reviewer 2 Report
Comments and Suggestions for Authors
The authors have presented a study on the ShK domain-containing protein in the nematode B.xylophius, shedding light on plant-parasitic interactions in tree species. They, for the first time, reported the involvement of the ShK protein in oxidative stress and suggested a role for reactive oxygen activity in host plants during parasitism. This study holds significance and is likely to be of interest to the research community.
However, I have identified some issues that need to be addressed to enhance the professionalism and acceptability of the manuscript.
Figure 4:
No control image is provided, such as those used in site hybridization employing no probe or sense Dig-labeled probes.
Figure 5:
Is the relative gene expression related to 0mM? I guess, but the presentation is unclear.
A clustered column chart is suggested to be used to compare values across different categories, such as different mM of hydrogen peroxide treatment.
Addressing these issues will contribute to the overall quality and credibility of the research findings
Author Response
Dear Reviewer 2,
We enclose a revised version of the manuscript entitled” Functional characterization of ShK domain-containing protein in the plant-parasitic nematode Bursaphelenchus xylophilus” which has been adjusted to take count of the reviewers comments.
We have carefully taken into consideration all comments and modified the manuscript text and figures following your suggestions. The responses are below.
Figure 4: No control image is provided, such as those used in situ hybridization employing no probe or sense Dig-labeled probes.
Response: The controle image is provided in figure 4 in image H (in the submitted manuscript). To make it more clear, we have added in the figure legend:”(H) sense probe (control)”.(lines 235-236)
Figure 5: Is the relative gene expression related to 0mM? I guess, but the presentation is unclear.
Response: The sentence was rephrased to: “Normalized expression ratio (log2) of the nine candidate genes and Bxy-ctl-1 (BXY_1386500) and Bxy-ctl-2 (BXY_1745800) under oxidative stress conditions, 15mM and 30mM H2O2, compared to control (0mM).” (lines 261-265)
A clustered column chart is suggested to be used to compare values across different categories, such as different mM of hydrogen peroxide treatment.
Response: Figure 5 has been changed to a new clustered column chart.
Round 2
Reviewer 1 Report
Comments and Suggestions for Authors
Dear Authors,
Thank you for taking the comments into consideration.
Please double check these minor final comments:
Line 189-190: use this sentence instead: “The multiple sequence alignment revealed that ShKT D1 domain sequences were more similar among each other than ShKT D2 domain. Same pattern was also observed within ShKT D2 domain sequences.”
Line 191: the sentence is not correct, rephrase it
Line 342: please us “phylogenetic analysis on B. xylophilus ShK domain-containing protein sequences” as a title
Good luck,
Reviewer
Author Response
Dear Reviewer,
We enclose a revised version of the manuscript entitled” Functional characterization of ShK domain-containing protein in the plant-parasitic nematode Bursaphelenchus xylophilus” which has been adjusted to take count of the reviewers comments.
We have carefully taken into consideration your comments. The responses are below.
Line 189-190: use this sentence instead: “The multiple sequence alignment revealed that ShKT D1 domain sequences were more similar among each other than ShKT D2 domain. Same pattern was also observed within ShKT D2 domain sequences.”
Response: The sentence was changed to “The multiple sequence alignment revealed that ShKT D1 domain sequences were more similar among each other than ShKT D2 domain. Same pattern was also observed within ShKT D2 domain sequences.”
Line 191: the sentence is not correct, rephrase it
Response: We have made edits to the text that should clarify the sentence: “The ML tree showed two clades: one with ShKT domains sequences of plants, considered as outgroup (A. suecica KAG7546197 and Eucalyptus grandis XP_010033232); and the other clade with ShKT domains sequences from Nematoda. In this last clade several sub-clades are formed. The first sub-clade is composed with animal-parasitic species B. malayi (VIO92324) and T. canis (KHN87551), and free-living nematodes B. okinawaensis (CAD5214648) and Diploscapter pachys (PAV73547) (domain 1). The second sub-clade presents several clusters, from which the first is formed with the domain 2 of the same species previously showed in the first sub-clade.”
Line 342: please us “phylogenetic analysis on B. xylophilus ShK domain-containing protein sequences” as a title
Response: As suggested in the first round of revisions of R1, we changed to “phylogeny analysis”. Now, as requested by this reviewer, the title was changed to “phylogenetic analysis on B. xylophilus ShK domain-containing protein sequences”.
Reviewer 2 Report
Comments and Suggestions for Authors
This revised version has been improved, and I believe it meets the criteria for publication acceptance.
Author Response
Dear Reviewer 2,
We kindly thank you for all your comments to our manuscript.
On behalf of the authors
Margarida espada